# ADAPTIVE GUIDANCE ACCELERATES REINFORCEMENT LEARNING OF REASONING MODELS

## ABSTRACT

We study the process through which reasoning models trained with reinforcement learning on verifiable rewards (RLVR) can learn to solve new problems. We find that RLVR drives performance in two main ways: (1) by compressing pass@$k$ into pass@1 and (2) via "capability gain" in which models learn to solve new problems that they previously could not solve even at high $k$. We find that while capability gain exists across model scales, learning to solve new problems is primarily driven through self-distillation. We demonstrate these findings across model scales ranging from 0.5B to 72B parameters on >500,000 reasoning problems with prompts and verifiable final answers across math, science, and code domains. We further show that we can significantly improve pass@$k$ rates by leveraging *natural language guidance* for the model to consider within context while still requiring the model to derive a solution chain from scratch. Based of these insights, we derive Guide – a new class of online training algorithms. Guide adaptively incorporates hints into the model's context on problems for which all rollouts were initially incorrect and adjusts the importance sampling ratio for the "off-policy" trajectories in order to optimize the policy for contexts in which the hints are no longer present. We describe variants of Guide for GRPO and PPO and empirically show that Guide-GRPO on 7B and 32B parameter models improves generalization over its vanilla counterpart with up to 4% macro-average improvement across math benchmarks. We include careful ablations to analyze Guide's components and theoretically analyze Guide's learning efficiency[1].

## 1 INTRODUCTION

Leading reasoning models on math, science, and coding benchmarks learn to utilize chain-of-thought via reinforcement learning with verifiable rewards (RLVR) (1; 2; 3; 4). These models are optimized to maximize verifiable rewards by comparing predicted final answers to ground truth. Models trained with RLVR are capable of surpassing previous approaches (such as SFT or RLHF) on challenging math and science benchmarks due to availability of verifiable rewards at scale. Yet the drivers of these gains—and how they evolve with model scale—remain poorly understood. Yue et al. (5) attribute RLVR's improvements almost entirely to the distillation of the base model's existing knowledge. In this work, we instead formalize RLVR's improvements as a sum of two orthogonal effects—distillation and genuine capability gain—and investigate how each of these effects evolves as models scale.

Specifically, there are at least two ways to improve a language model's ability to solve challenging reasoning problems autonomously:

1. By distilling knowledge from pass@$k$ into pass@1 (6; 7; 8; 9; 10; 11)

2. Capability gain via RL in which a language model learns to solve new problems it previously was not able to solve even when given $k$ attempts.

In this work, we propose a formalism to measure the extent to which learning during RLVR is driven by self-distillation or capability gain. We then seek to leverage these insights to accelerate learning

---

[1]Code will be open sourced.

of new problems during RLVR by incorporating guidance into the reasoning model's context. We therefore address two main research questions:

1. **Self-distillation or capability gain?** To what extent is learning during RLVR merely redistributing probability mass among outputs the model already knows ("self-distillation") versus genuinely expanding the model's problem-solving capabilities?

2. **Do guidance-conditioned trajectories on failure accelerate learning?** If we give the policy selective guidance on complete problem failure, while requiring the trajectories to be generated by the same policy state (and therefore close to the on-policy distribution), can we close knowledge gaps faster than (a) using fully off-policy data, (b) providing no guidance at all, or (c) always providing guidance?

Addressing these questions, our study yields three key contributions. First, we show that **improvements during RLVR are primarily driven by self-distillation**: models learn to compress pass@$k$ into pass@1 by shifting probability mass toward answers they could already reach with multiple attempts. Second, we find that **pass@$k$ itself can be significantly improved through selective guidance**: when the model fails all $k$ attempts, providing a hint in-context—while still requiring it to derive the reasoning chain from scratch—helps it discover successful trajectories that remain unreachable through naive sampling. Third, synthesizing these insights, **we introduce Guide**, a training algorithm that uses guided rollouts on failure to increase pass@$k$, thereby expanding the pool of answers available for self-distillation. Guide accelerates learning in RLVR by turning unreachable solutions into reachable ones, and by carefully correcting the importance sampling ratio, we enable the model to subsequently learn them without guidance. We validate Guide across math benchmarks and provide theoretical and empirical analysis of its learning efficiency.

## 2 METHODS

### 2.1 SELF-DISTILLATION VS. CAPABILITY GAIN

We study the post-training dynamics that govern LLMs learning to solve new tasks. We measure this ability as the rewards $\mathcal{R}$ acquired from an environment, such as the test set of a benchmark. Specifically, we are interested in how an LLM learns to solve *new* problems during RL. To this end, we define $\mathcal{R}^{net}$ as the sum of *net new* rewards acquired *after* RL for a policy $\pi_{\text{RL}}$

$$\mathcal{R}^{\text{net}} = \underbrace{\sum_{i \in U^{\pi_{\text{init}}}} \mathbb{I}[\hat{y}_i^{\pi_{\text{RL}}} = y_i]}_{progress} - \underbrace{\sum_{j \in S^{\pi_{\text{init}}}} \mathbb{I}[\hat{y}_j^{\pi_{\text{RL}}} \neq y_j]}_{regression} \tag{1}$$

where $U^{\pi_{\text{init}}}$ is the set of indices of unsolved problems prior to RL and $S^{\pi_{\text{init}}}$ is the set of indices of solved problems prior to RL. We define solved and unsolved here via pass@1 correctness. Note that $\mathcal{R}^{net}$ can be calculated against both training data and test data and in practice is equal to the change in accuracy before and after training.

Note that the progress term can be decomposed into problems that have at least one correct solution in a sample $\mathcal{Y}_i = \{\hat{y}_1, \ldots, \hat{y}_k\}$ of $k$ responses from $\pi_{\text{init}}$ to the same prompt (i.e. pass@$k$ = 1) and problems that have no correct solutions in the sample (i.e. pass@$k$ = 0).

$$\underbrace{\sum_{i \in U^{\pi_{\text{init}}}} \mathbb{I}[\hat{y}_i^{\pi_{\text{RL}}} = y_i]}_{progress} = \underbrace{\sum_{i \in U^{\pi_{\text{init}}}} \mathbb{I}\left[\exists\, \hat{y} \in \mathcal{Y}_i \text{ s.t. } \hat{y} = y_i \wedge \hat{y}_i^{\pi_{\text{RL}}} = y_i\right]}_{distillation} + \underbrace{\sum_{i \in U^{\pi_{\text{init}}}} \mathbb{I}\left[\forall\, \hat{y} \in \mathcal{Y}_i, \hat{y} \neq y_i \wedge \hat{y}_i^{\pi_{\text{RL}}} = y_i\right]}_{capability\ gain}$$

In order to understand how RLVR teaches models to solve new reasoning problems in practice, we set $k$ equal to the number of rollouts per problem used during training ($k$ may be set higher and we define effective vs. absolute capability gain in Appendix §C).

Decomposing progress into the above terms enables us to understand the mechanisms driving RLVR. We empirically analyze these components in Section 3.1 and find that while effective capability gain exists, progress is dominated by self-distillation.

---

**Algorithm 1 Guide-GRPO**: GRPO with guidance-augmented rollouts on failure

---

**Input:** initial policy $\pi_{\theta_{\text{init}}}$; reward model $r_\varphi$; task prompts $\mathcal{D}$; hyper-parameters $\varepsilon$, $\beta$, $\mu$, $k$ roll-outs per prompt, guidance suffix guid

1:   $\pi_\theta \leftarrow \pi_{\theta_{\text{init}}}$
2: **for** $iter = 1, \dots, I$ **do**
3:     $\pi_{\text{ref}} \leftarrow \pi_\theta$                                                    ▷ freeze reference
4:     **for** $step = 1, \dots, M$ **do**
5:        Sample minibatch $\mathcal{D}_b \subset \mathcal{D}$
6:        $\pi_{\theta_{\text{old}}} \leftarrow \pi_\theta$                                      ▷ snapshot old policy
7:        Sample $K$ outputs $\{o_i\}_{i=1}^K \sim \pi_{\theta_{\text{old}}}(\cdot \mid q)$ for every $q \in \mathcal{D}_b$
8:        Identify unsolved set $U = \{\, q \in \mathcal{D}_b : \textit{all } k \textit{ roll-outs fail}\,\}$
9:        **for** $q \in U$ **do**
10:          Sample $k$ guidance rollouts $\tilde{o} \sim \pi_{\theta_{\text{old}}}(\cdot \mid \langle q, \texttt{guid} \rangle)$
11:        **end for**
12:        Compute rewards $r_i = r_\varphi(o_i)$ (and $r_{\tilde{o}}$ if present)
13:        Compute advantages $\hat{A}_{i,t}$ via group-relative estimation
14:        **for** $gstep = 1, \dots, \mu$ **do**
15:          Update $\pi_\theta$ by maximising the Guide objective in Eq. 2.2
16:        **end for**
17:     **end for**
18: **end for**
19: **return** $\pi_\theta$

**Output:** fine-tuned policy $\pi_\theta$

---

## 2.2   GUIDE: ACCELERATING LEARNING WITH GUIDANCE ON FAILURE

Inspired by our empirical results showing that self-distillation dominates learning of new problems during RLVR (see Figure 1), concurrent work showing similar results (5), and a rich history of success in RL of using off-policy data to improve training efficiency (12), we seek to increase the proportion of correct rollouts during RL. We hypothesize that a particularly effective means to do this will be by *guiding* the policy with a prompt-specifc hint, $h$, such that the model is required to reach the solution in its own terms: $\pi_\theta(o_{i,t} \mid q, h, o_{i,<t})$. In an initial validation of this hypothesis, we find that including hints significantly improves pass@k, as shown in Figure 2. To this end, we derive a new class of online RL training algorithms which we call Guide. We describe the general form and a specialization to PPO in Appendix §A. Further, we carefully analyze a specialization of Guide to GRPO in which we (1) provide guidance on unsolved prompts and (2) apply an off-policy importance weight so that samples drawn with guidance still optimize performance *without* guidance, as shown in Algorithm 1.

**GRPO**   In typical RLVR with GRPO, for each question $q$, we sample $k$ outputs $\{o_i\}_{i=1}^k$ from the old policy $\pi_{\theta_{\text{old}}}(\cdot \mid q)$ and score them, yielding rewards $\{r_i\}_{i=1}^k$. We apply per-prompt $z$-normalization and set the token-level advantages $\hat{A}_{i,t}$ for all tokens $t$ in each output $o_i$ equal to the corresponding normalized reward $\hat{A}_{i,t} = \tilde{r}_i = \frac{r_i - \mu_r}{\sigma_r}, t = 1, \dots, |o_i|$.

The GRPO objective maximized during policy updates is defined as:

$$\mathcal{J}_{\text{GRPO}}(\theta) = \mathbb{E}\left[\min\left\{\frac{\pi_\theta}{\pi_{\theta_{\text{old}}}} \hat{A}_{i,t}, \text{clip}\left(\frac{\pi_\theta}{\pi_{\theta_{\text{old}}}}, 1-\varepsilon, 1+\varepsilon\right)\hat{A}_{i,t}\right\} - \beta D_{KL}\right] \tag{2}$$

where $\varepsilon$ and $\beta$ are hyperparameters controlling clipping and KL regularization, respectively.

**Guide**   We make the observation that because we want the model to perform well without guidance, the guided trajectories are off-policy. To avoid biasing the gradient, we should appropriately compute the importance weight (Sutton & Barto 1998). To this end, we modify the GRPO objective to

$$\mathcal{J}_{\text{Guide}}(\theta) = \mathbb{E}_{\mathcal{S}(q)}\left[\left\{\min\left[\underbrace{\frac{\pi_\theta(r_t \mid x_q, r_{<t})}{\pi_{\theta_{\text{old}}}(r_t \mid s_q, r_{<t})}}_{\text{importance weight}}\hat{A}_{r,t}, \text{clip}\left(\underbrace{\frac{\pi_\theta(r_t \mid x_q, r_{<t})}{\pi_{\theta_{\text{old}}}(r_t \mid s_q, r_{<t})}}_{\text{importance weight}}, 1 \pm \varepsilon\right)\hat{A}_{r,t}\right] - \beta\, D_{KL}\right\}\right]$$

where $\mathcal{S}(q)$ is the set of $k$ sampled rollouts for prompt $q$, containing $k$ plain rollouts $r \sim \pi_{\theta_{\text{old}}}(\cdot \mid x_q)$ and, if all fail, $k$ guided rollouts $r \sim \pi_{\theta_{\text{old}}}(\cdot \mid \tilde{x}_q)$, where $x_q$ and $\tilde{x}_q$ are the plain and guided prompts respectively, $s_q \in \{x_q, \tilde{x}_q\}$ is the prompt used to generate rollout $r$, $\hat{A}_{r,t}$ is the group-normalized advantage at token $t$, and $\varepsilon, \beta$ are the PPO-style clipping and KL-regularization hyperparameters. Guide injects hints only when all unguided rollouts fail, and an importance weight projects those off-policy trajectories onto the on-policy gradient direction. This focuses learning signal on the hardest unsolved problems while keeping every guided update aligned with the plain-prompt objective, thereby achieving faster progress than vanilla GRPO. We formalize this notion into the following theorem and provide a proof in Appendix §B:

**Theorem 1 (Guide-GRPO improves learning efficiency)** *Let $U$ be the set of prompts $q$ unsolved by the current policy $\pi_\theta$. Suppose that, in expectation over unsolved prompts and the group $G_q$ of guided and unguided trajectories, the guided advantage is positive:*

$$\mathbb{E}_{q \in U}\left[ \mathbb{E}_{y \sim \pi_\theta(\cdot | \tilde{x}_q)}\left[ \tilde{A}_q(y; G_q) \right] \right] > 0.$$

*Then for all $\eta$ sufficiently small, the one-step expected improvement, $\Delta \mathcal{R}$, under Guide-GRPO exceeds that of Vanilla GRPO, to first order in $\eta$:*

$$\mathbb{E}[\Delta \mathcal{R}_{\text{Guide}}] > \mathbb{E}[\Delta \mathcal{R}_{\text{Vanilla}}], \tag{3}$$

*where*

$$\mathbb{E}[\Delta \mathcal{R}_{\text{Vanilla}}] = \eta \sum_{q \in U} A_q \, p_q^2 + \mathcal{O}(\eta^2), \tag{4}$$

$$\mathbb{E}[\Delta \mathcal{R}_{\text{Guide}}] = \eta \sum_{q \in U} \left[ A_q \, p_q^2 + (1 - p_q)^k \, \mathbb{E}_{y \sim \pi_\theta(\cdot | \tilde{x}_q)}[\tilde{A}_q(y)] \, p_q \right] + \mathcal{O}(\eta^2), \tag{5}$$

*and $p_q = \mathbb{P}_{y \sim \pi_\theta(\cdot | x_q)}\left[ f(y) = y_q^* \right]$ denotes the success probability under the unguided policy.*

Note that Guide's relative gain over vanilla GRPO increases when

- *failure probability* $(1 - p_q)^k$ is large (hard prompts),
- *guided advantage* $\mathbb{E}[\tilde{A}_q]$ is large on average relative to the full rollout group,
- the success probability under the unguided policy $p_q$ is non-zero (so credit can propagate).

## 3 EXPERIMENTS

### 3.1 RLVR DRIVES LEARNING PROGRESS MAINLY VIA SELF-DISTILLATION

We investigate the mechanisms driving performance improvements in models trained using RLVR, explicitly decomposing the observed improvements into two measurable effects: capability gains and self-distillation. Concretely, for the experiments in this section, we define capability gain and self-distillation as follows:

**Capability gain** The count of problems that are initially unsolved by the untrained policy, even with multiple attempts `pass@16`, which subsequently become solvable by the RLVR-trained policy within a single sample (`pass@1`). [2]

**Self-distillation** The count of problems solvable by the untrained policy with multiple sampling attempts (`pass@16`) that later become solvable with just one attempt (`pass@1`) during RLVR training.

---

[2] In practice, we compute the unsolved set $U$ using pass@1 & temperature 0, the *potential* distill set $D$ with pass@16 & temperature 1 (to mirror training settings), and *potential* capability gain set $G = U - D$. Solved problems are defined by pass@1 temperature 0.

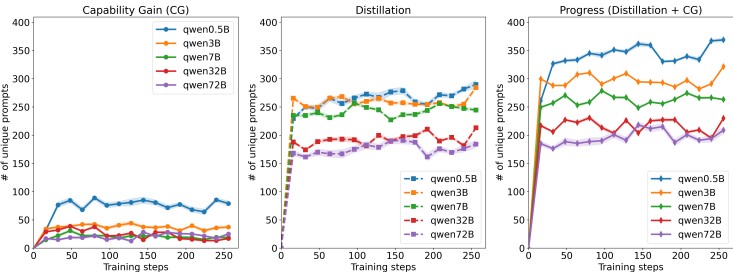

Figure 1: Capability gain (left), self-distillation (middle), and combined progress (capability gain + self-distillation; right) across training steps on all test sets.

### 3.1.1 EXPERIMENTAL SETUP

For our base models, we use Qwen 2.5 (13) at five model scales, 0.5B, 3B 7B, 32B, and 72B as the starting untrained policies. Each run is trained for 256 steps using the GRPO training objective on a dataset composed of math, stem, and coding tasks. We evaluate every 16 training steps on the following benchmarks: GSM8K (14), MATH500 (15), AIME24 (16), AIME25 (17), AMC23 (18), GPQA-DIAMOND (19), OLYMPIADBENCH (20), LEETCODE (21), LIVECODEBENCH (22), and HUMANEVAL (23). To measure variance in capability gain and self-distillation across runs (as defined in 3.1), we perform 10 independent trials, each with its own random seed. We first generate 100 rollouts at temperatures 1.0 and 0.0 for every problem in the full test set. Then for each trial, to compute pass@1, we randomly sampling one of the 100 temperature 0.0 rollouts and judge its correctness; to compute pass@16, we randomly sample 16 trajectories from the temperature 1.0 rollouts and judge if any of the sample are correct. We apply this sampling procedure independently across the 10 trials and aggregate results to report the overall mean and standard error of capability gain, distillation, and progress counts on the full test set. Additional training hyper-parameters and implementation details are provided in Appendix § H.

### 3.1.2 ANALYSIS

Figure 1 decomposes net performance gains (Eq. 2.1) into capability gain and self-distillation. We make the following observations:

**Self-distillation dominates**    Across all four Qwen sizes, the majority of the progress improvement comes from converting answers that were already reachable within $\leq 16$ untrained samples into the trained pass@1 at temp 0. Among the models evaluated, Qwen 7B and Qwen 3B shows the highest gain via self-distillation, whereas the larger models (Qwen 32B and Qwen 72B) showed comparatively fewer gains from self-distillation. In contrast, every model learns to solve some problems it could not solve at initialization, with the 0.5B model gaining the most in relative terms. Nevertheless, capability gain remains a minority contributor at every scale, indicating that RLVR primarily re-allocates probability mass rather than discover truly novel solutions at the studied $k$.

**Headroom dictates returns and shrinks with capability**    We first note that the unsolved set $|U|$ contracts sharply as model size grows: 0.5B begins with 3195 unsolved items, 3B with 2150, 7B with 1913, 32B with 1617, and 72B with just 1532. Because each model converts a similar fraction of its own $|U|$ ($\approx 25\%$), the absolute count of pass@1 lift (Figure 1; right) inevitably drops at larger scale. Progress for stronger models therefore hinges on introducing harder examples that replenish $|U|$ and expose new reasoning gaps.

### 3.2 GUIDE-GRPO TOWARDS MATHEMATICAL REASONING

Leveraging the observation that the majority of the performance gain in RLVR training is from self-distillation, we seek to increase the proportion of correct trajectories during RL training while remaining close to the policy's sampling distribution. In this section, we first validate the hypothesis that prompt-specific guidance in the model's context improves pass@k (Figure 2), and then utilize this improvement to empirically demonstrate Guide-GRPO's (Algorithm 1) effectiveness towards improving mathematical reasoning for language policy models (Table 1 and Table 2).

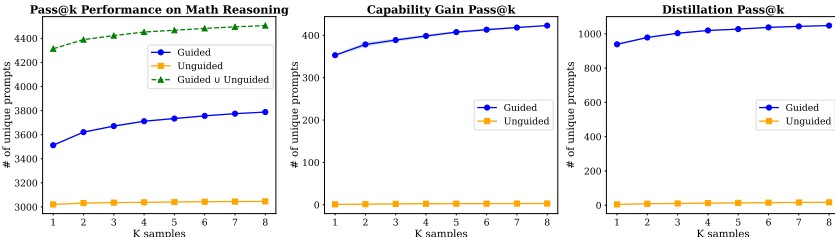

Figure 2: Impacts of guidance on correct rollouts. **Left**: Guidance vs. no-guidance pass@k performance on Qwen-2.5-Math-7B on 10K randomly sampled training examples from open-r1/OpenR1-Math-220k (24). Including problem-specific guidance into the context increases unbiased pass@$k$. **Middle**: Guided rollouts solve more previously unsolvable questions (capability gain), with gains growing in k. **Right**: Guidance also improves performance on the distillation subset in comparison to unguided model.

### 3.2.1 EXPERIMENTAL SETUP

Our training data consists of the default subset of OpenR1-Math-220k (24), comprising 93.7K math reasoning tasks. For each entry, we extract the prompt, ground-truth answer, and the human-authored reference solution. For guidance generation, we prompt GPT-4o to produce pedagogically-inspired hints that mimic expert tutoring strategies – providing high-level conceptual direction and problem-solving frameworks without revealing solution paths (full instructions and guidance examples are in Appendix §G).

Our base model, Qwen-2.5-Math-7b (13), is a large language model pre-trained and fine-tuned for complex mathematical reasoning. We establish a comprehensive comparative framework: (1) standard GRPO training, (2) GRPO with Filtering – a technique shown to improve training efficiency by discarding prompts for which the rollouts are all incorrect or all correct (25), (3) our proposed Guide-GRPO approach, (4) a SFT baseline trained directly on human-authored solutions, (5) expert iteration (STaR) and (6) context distillation (6; 26). This multi-faceted comparison allows us to evaluate whether transforming expert solutions into guided hints yields performance advantages over both direct imitation of expert reasoning and optimized reinforcement learning approaches. Moreover, to assess the robustness of our method to increasing computational resources, we conduct experiments that increase context length (4K → 8K) followed by an increase in context length *and* model size (7B → 32B). Additional training hyper-parameters and implementation details are provided in Appendix §I. We also include results showing similar improvements on Llama-3.1-8B Instruct, which can found in Table 6 in the Appendix.

### 3.2.2 RESULTS

**Task-specific guidance increases correct rollouts**    Figure 2 demonstrates that introducing targeted, in-context guidance significantly increases the number of correct rollouts. We then dissect how these hints affect both capability gain and self-distillation (see Section 3.1). Our analysis reveals that guidance not only helps the model solve previously unreachable prompts (capability gain) but also reinforces consistency on already-solvable ones (self-distillation). Building on this insight, we apply Guide-GRPO (see Section 3.2) to transfer performance improvements observed under guidance to directly improve the base policy. More details about this experimental setup are described in Appendix I.

**Guide-GRPO leads to better test-time performance**    As shown in Table 1, Guide-GRPO consistently outperforms all baselines across both pass@1 and pass@16 metrics on a wide range of math benchmarks. Notably, Guide-GRPO achieves a 3% absolute improvement in pass@1 on Olympiad-level questions and a 13% improvement in pass@16 on AIME 25, relative to the next best performing baseline. On aggregate, it achieves the highest macro-average (51.03 pass@1, 70.15 pass@16) and micro-average (70.66 pass@1, 83.29 pass@16) scores, highlighting robust gains across both balanced and volume-weighted evaluations.

Table 1: Comparison of Pass@1 (greedy decoding) and Pass@16 (temperature 1.0) performance on several math benchmarks across different training algorithms. SFT is trained on the reference solution, Filter-GRPO uses the standard GRPO objective with filtering of all incorrect and all correct groups, GRPO is without filtering, Base is the base model (Qwen-2.5-Math-7B), STaR is expert iteration, Ctx Dist. is context distillation, and Guide-GRPO is our method. The performance for Pass@1 is averaged over 5 independent samples. Table 4 contains the full results with 95% confidence intervals. Bold values indicate best performance.

| Benchmark | Metric | Guide | Filter-GRPO | GRPO | STaR | Ctx Dist. | SFT | Base |
|---|---|---|---|---|---|---|---|---|
| MATH500 | P@1 | **82.68** | 80.80 | 79.00 | 77.20 | 76.92 | 72.80 | 68.80 |
|  | P@16 | **93.60** | 92.60 | 90.80 | 91.20 | 90.40 | 87.00 | 89.60 |
| GSM8K | P@1 | 91.43 | **91.84** | 91.71 | 87.20 | 88.20 | 88.22 | 83.21 |
|  | P@16 | **97.73** | 96.59 | 96.97 | 98.10 | 98.18 | 96.89 | 97.19 |
| MINERVA | P@1 | 32.35 | 30.51 | **32.72** | 31.03 | 27.94 | 26.25 | 26.47 |
|  | P@16 | **47.43** | 43.75 | 45.96 | 48.16 | 41.91 | 35.29 | 43.01 |
| OLYMPIAD | P@1 | **43.11** | 40.21 | 39.56 | 36.10 | 35.41 | 35.05 | 33.07 |
|  | P@16 | **64.59** | 60.89 | 61.33 | 57.78 | 54.52 | 50.52 | 52.15 |
| AMC | P@1 | **63.61** | 60.24 | 62.41 | 44.58 | 49.16 | 50.36 | 47.47 |
|  | P@16 | **84.34** | **84.34** | **84.34** | 75.90 | 80.72 | 69.88 | 78.31 |
| AIME 24 | P@1 | **30.67** | 18.67 | 26.67 | 13.33 | 22.67 | 11.33 | 6.67 |
|  | P@16 | 56.67 | 53.33 | **60.00** | 40.00 | 43.33 | 23.33 | 33.33 |
| AIME 25 | P@1 | **13.33** | **13.33** | **13.33** | 6.67 | 15.33 | 6.67 | 3.33 |
|  | P@16 | **46.67** | 33.33 | 30.00 | 10.00 | 20.00 | 13.33 | 13.33 |
| Macro Avg. | P@1 | **51.03** | 47.94 | 49.34 | 42.30 | 44.76 | 40.42 | 38.43 |
|  | P@16 | **70.15** | 66.40 | 67.06 | 60.16 | 61.30 | 53.75 | 58.13 |
| Micro Avg. | P@1 | **70.66** | 69.76 | 69.59 | 65.52 | 65.62 | 63.80 | 61.16 |
|  | P@16 | **83.29** | 81.23 | 81.44 | 80.75 | 79.58 | 76.28 | 78.31 |

These results demonstrate that Guide-GRPO is effective at integrating prompt-specific guidance into the training process, enabling the resulting policy to generalize better to difficult mathematical reasoning task – even without access to guidance at test time. Additionally, its strong pass@k performance, combined with our observation that RLVR primarily drives progress through self-distillation, suggests that Guide-GRPO promotes better exploration and solution diversity, which are key to continued improvement in reasoning-centric domains.

**Guide-GRPO improvements scale with context length and model size**   The results in Table 2 demonstrates Guide-GRPO's consistent improvements over vanilla GRPO when scaling to larger context lengths (4K → 8K) and model sizes (7B → 32B). For the 32B model, Guide-GRPO achieves 3.39 percentage point improvement in macro-average Pass@1 (56.26% vs 52.87%) and 1.89 percentage point improvement in micro-average Pass@1 (76.36% vs 74.47%). More generally, the improvements are consistent across both Pass@1 and Pass@16 metrics, with Guide-GRPO showing gains ranging from 1-4 percentage points across all configurations. These results strengthen the empirical evidence that Guide-GRPO's test-time generalization scales effectively with increased computational resources along both context length and parameter count dimensions.

Table 2: Comparison of Pass@1 (greedy decoding) and Pass@16 (temperature 1.0) performance on several math benchmarks with larger context length (8K) across model sizes (7B and 32B). The performance for Pass@1 is averaged over 5 independent samples. Table 5 contains the full results with 95% confidence intervals. Bold values indicate best performance.

| Benchmark | Metric | Guide-32B-8K | GRPO-32B-8K | Guide-7B-8K | GRPO-7B-8K |
|---|---|---|---|---|---|
| Macro Avg. | P@1 | **56.26** | 52.87 | 49.58 | 49.29 |
|  | P@16 | **74.08** | 71.84 | 72.15 | 68.42 |
| Micro Avg. | P@1 | **76.36** | 74.47 | 71.15 | 69.89 |
|  | P@16 | **85.63** | 84.94 | 84.29 | 83.84 |

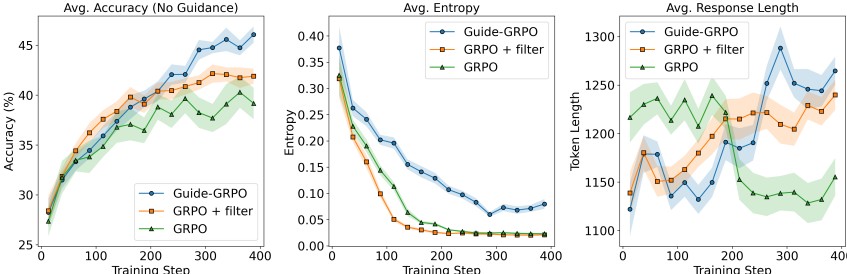

Figure 3: Comparison of Guide-GRPO with baseline methods across training steps (400 total). Left: Rollout accuracy without guidance shows Guide-GRPO ultimately outperforming baselines despite lower initial performance. Middle: Generation entropy remains consistently higher for Guide-GRPO, indicating better solution diversity. Right: Response token length increases for Guide-GRPO in later training stages. Shaded regions represent confidence intervals.

**Guide-GRPO demonstrates better train-time metrics** Figure 3 reveals an interesting training trajectory for Guide-GRPO. While initially exhibiting lower rollout accuracy without guidance, Guide-GRPO ultimately surpasses standard GRPO methods as training progresses. This performance crossover indicates our selective guidance injection approach effectively updates policy weights, enabling the model to perform better independently without requiring guidance at inference time. Notably, Guide-GRPO maintains consistently higher entropy throughout training while steadily increasing response length. This combination of enhanced entropy and improved performance, both during training and testing, suggests that Guide-GRPO preserves exploratory capacity for novel solutions while achieving superior results across diverse mathematical reasoning tasks.

**Training dynamics reveal critical convergence factors for Guide-GRPO** Our investigation into various policy loss formulations uncovered specific configurations that lead to consistent training instability. Figure 4 in Appendix §D illustrates the reward trajectories across different settings, highlighting two critical factors affecting convergence:

- **Importance weighting relative to guided distribution** – Constructing importance weights for guided trajectories relative to old policy weights conditioned solely on the prompt introduces significant training instability. Since the sampled trajectories originate from the old policy conditioned on both prompt and guidance—rather than just the prompt—the resulting probability ratios between current and old policy weights misrepresent the true gradient direction along the sampled trajectory, leading to suboptimal updates. A theoretical support is detailed in Appendix §B.

- **PPO-Clip mechanism destabilizes guided trajectories** – When incorporating guided trajectories with importance weighting relative to the sampling distribution, we observe that PPO-clipping causes training divergence at approximately 50 steps. This phenomenon aligns with theoretical expectations: guided trajectories inherently generate smaller probability ratios, causing the minimum clip operation to artificially inflate most token probability ratios, thereby triggering unstable gradient updates. We mitigated this issue by removing ratio clipping, which empirically produced stable training outcomes.

**Threshold for guidance** Our ablation across three guidance thresholds (All Incorrect, Mostly Incorrect, and Always) reveals optimal performance when guidance is applied only when all standard rollouts fail, as shown in Table 3. While "Mostly Incorrect" performs comparably, unconditional guidance significantly impairs results. Excessive guidance handicaps learning by preventing the model from developing robust reasoning. Conversely, strategic guidance only for entirely incorrect samples provide essential signal when the model's sampling distribution completely misses valid solutions, providing exposure to guided solution traces to problems beyond the current policy's capability while incentivizing independent exploration in all other cases.

Table 3: Performance comparison of various guidance threshold strategies. The "All Incorrect" strategy applies guidance only when all original prompt rollouts fail; "Mostly Incorrect" applies guidance when accuracy falls below 25%; and "Always" unconditionally applies guidance to all rollouts. Bold values indicate best performance.

|  | Metric | All Incorrect | Mostly Incorrect | Always |
|---|---|---|---|---|
| Macro Avg. | P@1 | **51.03** | 50.48 | 40.97 |
|  | P@16 | **70.15** | 69.86 | 60.03 |
| Micro Avg. | P@1 | **70.66** | 70.14 | 63.18 |
|  | P@16 | **83.29** | **83.29** | 80.72 |

## 4 RELATED WORK

**Reinforcement Learning for LLM Reasoning**   Recent advances in reinforcement learning approaches (27; 28; 29; 30; 31) have demonstrated remarkable progress in enhancing LLMs' reasoning capabilities. OpenAI-o1 (1) and DeepSeek-R1 (2) have generated state-of-the-art results in complex reasoning tasks such as in math, coding, etc. by pioneering the use of Reinforcement Learning from Verifiable Rewards (RLVR) (3; 32; 2; 1), in which the reward is computed using a rule-based verification function (33; 34; 35). Some works also provide frameworks for the distillation of knowledge from pass@$k$ into pass@1 via expert iteration (6; 7; 8; 9; 10; 11) to improve a language model's ability to solve challenging reasoning problems autonomously.

**Learning Mechanisms for Reinforcement from Verifiable Rewards**   Building upon the increasing traction of RLVR in the reasoning space, some works have addressed the fundamental dynamics of improvements seen from RLVR (5; 36; 37), claiming that RLVR boosts sampling efficiency by biasing the model's output distribution toward paths that are more likely to yield rewards but reduces the overall reasoning capacity boundary at very high $k$ ($= 256$) (5). We corroborate the results regarding sampling efficiency in our work as well, revealing that while capability gain exists in the lower $k$ range ($<= 16$) across multiple domains (math, coding, STEM, etc.) and model scales, learning to solve new problems via RLVR is dominated by self-distillation of pass@k performance into pass@1 performance.

**Reinforcement Learning for LLMs with Off-Policy Data**   A variety of off-policy reinforcement learning techniques such as DPO (38) and variants of on-policy algorithms like Tapered Off-Policy REINFORCE (39) have been applied to LLMs recently. Off-policy methods yield the advantage of enabling better sample efficiency by learning from experiences collected by different policies, but at the cost of potential increased instability. One recent work (40) targets integrating high-quality off-policy trajectories with policy shaping via regularized importance sampling. In contrast, our method leverages guidance in context, which we hypothesize has the potential to bridge the gap in benefits between on-policy and off-policy learning than fully off-policy incorporation and can be applied to training settings in which no more power teacher model exists to distill from. Thus, we focused on studying how to improve model performance independent of directly distilling from a much stronger model such as R1.

## 5 FUTURE WORK

While Guide-GRPO demonstrates strong empirical and theoretical performance on mathematical reasoning, several directions remain open. First, future work should more deeply investigate the effect of the quality and nature of guidance on model progress during RL. Future methods may dynamically generate guidance targeted at specific reasoning failures in the policy's trajectories within a multi-agent RL setting. Extending Guide to other domains such as code generation, agents, or even robotics could test its generality. In these initial experiments we have only evaluated Guide on models at 32B-parameter scale and at context lengths up to 8k due to compute limitations. Scaling studies are needed to understand how the effectiveness of Guide varies with model size, context length, and compute scale.

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

# A  GUIDE ALGORITHMS

The Guide algorithm is a general method for adaptively incorporating guidance into online RL. The general form is given as:

---

**Algorithm 2** Guide

---

1: **for** iteration=1,2,... **do**
2:      **for** step=1,2,...,N **do**
3:          Run policy $\pi_{\theta_{\text{old}}}$ in environment for $T$ timesteps
4:      **end for**
5:      Identify unsolved set $U = \{q : \text{all rollouts fail}\}$
6:      For $q \in U$, sample guided rollouts $\tilde{o} \sim \pi_{\theta_{\text{old}}}(\cdot|\langle q, \texttt{guid}\rangle)$
7:      Compute unguided $\hat{A}_t$ and guided advantages $\hat{A}_{\tilde{o},t}$
8:      Optimize objective $\mathcal{J}$ in eq. 2.2 wrt $\theta$
9:      $\theta_{\text{old}} \leftarrow \theta$
10: **end for**

---

Guide can also be specialized to PPO, integrating selective hinting on failed prompts and importance-sampling corrections directly into PPO's training loop. We call this specialization **Guide-PPO**.

---

**Algorithm 3** Guide-PPO

---

**Input:** initial policy $\pi_{\theta_{\text{init}}}$; reward model $r_{\varphi}$; task prompts $\mathcal{D}$; hyperparameters $\beta, \gamma$.
1: **for** iteration = 1,2,... **do**
2:      $\pi_{\theta_{\text{old}}} \leftarrow \pi_{\theta}, \quad \pi_{\text{ref}} \leftarrow \pi_{\theta}$
3:      Sample minibatch $\mathcal{D}_b \subset \mathcal{D}$
4:      Sample $k$ rollouts $\{o_i\}_{i=1}^{k} \sim \pi_{\theta_{\text{old}}}(\cdot|q)$ for all $q \in \mathcal{D}_b$
5:      Compute rewards $r_i = r_{\varphi}(o_i)$ and estimate advantages $\hat{A}_{i,t}$ via GAE
6:      Identify unsolved set $U = \{q \in \mathcal{D}_b : \textit{all } k \text{ rollouts fail}\}$
7:      **for** $q \in U$ **do**
8:          Sample $k$ guidance rollouts $\tilde{o} \sim \pi_{\theta_{\text{old}}}(\cdot|\langle q, \texttt{guid}\rangle)$
9:          Compute guided rewards $r_{\tilde{o}} = r_{\varphi}(\tilde{o})$
10:         Recompute advantages $\hat{A}_{\tilde{o},t}$ combining original and guided rollouts
11:      **end for**
12:      Optimize objective $\mathcal{J}$ in eq. 2.2 wrt $\theta$
13: **end for**

---

Where:

- $\pi_{\theta}$: policy being trained; $\pi_{\theta_{\text{old}}}$: sampling policy from previous iteration.

- $\pi_{\text{ref}}$: fixed KL-penalty reference policy.

- $\mathcal{D}_b$: sampled minibatch of prompts.

- $U$: prompts where all original rollouts fail, triggering guided rollouts.

- $\tilde{o}$: rollouts sampled with additional guidance suffix $\texttt{guid}$.

- $\hat{A}_{i,t}$: advantages estimated via Generalized Advantage Estimation (GAE).

- $f(\cdot)$: shaping function correcting importance sampling.

- $\beta$: KL divergence regularization coefficient.

# B  LEARNING EFFICIENCY OF GUIDANCE-AUGMENTED GRPO

## B.1  PRELIMINARIES

Let $\pi_\theta(y \mid x)$ be an autoregressive language model that samples a trajectory of token outputs $y = (y_1, \ldots, y_T)$. The success probability $p_q$ is defined as the marginal probability of sampling a trajectory that leads to a correct final answer. Gradients and advantages refer to whole-sequence log-probabilities and returns unless otherwise noted.

For each prompt $q$ with plain input $x_q$ and guided input $\tilde{x}_q$, define

$$p_q(\theta) = \mathbb{P}_{y \sim \pi_\theta(\cdot \mid x_q)} \left[ f(y) = y_q^* \right], \qquad \tilde{p}_q(\theta) = \mathbb{P}_{y \sim \pi_\theta(\cdot \mid \tilde{x}_q)} \left[ f(y) = y_q^* \right] \tag{6}$$

the success probabilities without and with guidance, respectively. For compactness, we overload notation and write

$$p_q := p_q(\theta_t), \tag{7}$$

to denote the scalar success probability at time step $t$, while retaining $p_q(\cdot)$ to indicate its dependence on $\theta$ elsewhere in the analysis.

A vanilla GRPO update with step size $\eta > 0$ is

$$\theta_{t+1} = \theta_t + \eta \sum_{q \in U} A_q \nabla_\theta \log \pi_\theta(y_q \mid x_q), \tag{8}$$

where $A_q$ is a group-normalised scalar advantage, computed over both guided and unguided rollouts.

When *all* $k$ plain rollouts for $q$ fail, **Guide-GRPO** draws $k$ guided rollouts on $\tilde{x}_q$ and applies an importance weight

$$w_q = \frac{\pi_\theta(y_q \mid x_q)}{\pi_{\theta_{\text{old}}}(y_q \mid \tilde{x}_q)} > 0 \tag{9}$$

to make the guided gradient on-policy.

Let $U$ be the set of problems unsolved by the initial policy.

We assume that for each question $q \in U$, the guided outputs have positive expected advantage relative to the full group of both guided and unguided rollouts $G_q$, i.e.,

$$\mathbb{E}_{q \in U} \left[ \mathbb{E}_{y \sim \pi_\theta(\cdot \mid \tilde{x}_q)} \left[ \tilde{A}_q(y; G_q) \right] \right] > 0. \tag{10}$$

This allows for the possibility that guidance may not always help, but is beneficial on average. For brevity, we define:

$$\mathbb{E}[\tilde{A}_q] := \mathbb{E}_{y \sim \pi_\theta(\cdot \mid \tilde{x}_q)} \left[ \tilde{A}_q(y; G_q) \right], \tag{11}$$

## B.2  LEMMAS

**Lemma 1 (Importance sampling aligns gradients)**  *For any guided sample* $y_q \sim \pi_{\theta_{old}}(\cdot \mid \tilde{x}_q)$,

$$\nabla_\theta \log \pi_\theta(y_q \mid \tilde{x}_q) = w_q \nabla_\theta \log \pi_\theta(y_q \mid x_q) \tag{12}$$

*so the guided gradient is a* positive scalar multiple *of the plain gradient. Consequently,*

$$\cos\left(\nabla_\theta \log \pi_\theta(y_q \mid x_q), \, w_q \nabla_\theta \log \pi_\theta(y_q \mid x_q)\right) = 1. \tag{13}$$

**Lemma 2 (Selective guidance outperforms or matches always-guidance)**  *Let* $\tilde{A}_q^{fail}$ *and* $\tilde{A}_q^{succ}$ *denote the expected guided advantages conditioned on whether all $k$ plain rollouts fail or at least one succeeds, respectively:*

$$\tilde{A}_q^{fail} := \mathbb{E}_{y \sim \pi_\theta(\cdot \mid \tilde{x}_q)} \left[ \tilde{A}_q(y; G_q) \,\middle|\, \text{all } k \text{ plain rollouts fail} \right], \tag{14}$$

$$\tilde{A}_q^{succ} := \mathbb{E}_{y \sim \pi_\theta(\cdot \mid \tilde{x}_q)} \left[ \tilde{A}_q(y; G_q) \,\middle|\, \text{at least one plain success} \right]. \tag{15}$$

*Then the expected first-order improvement in the number of unguided solutions is greater for* selective *guidance than for* always *guidance:*

$$\Delta^{sel} \geq \Delta^{all}. \tag{16}$$

*Moreover, since $\tilde{A}_q(y; G_q)$ is computed relative to the full set of guided and unguided rollouts, and the group mean reward is lower when all plain rollouts fail, we have $\tilde{A}_q^{fail} \geq \tilde{A}_q^{succ}$ by construction.*

**Proof 1** *Selective guidance applies guided updates only when all $k$ plain rollouts fail, so:*

$$\Delta^{sel} = \eta \sum_{q \in U} \left[ A_q p_q^2 + (1 - p_q)^k \tilde{A}_q^{fail} p_q \right], \tag{17}$$

$$\Delta^{all} = \eta \sum_{q \in U} \left[ A_q p_q^2 + (1 - p_q)^k \tilde{A}_q^{fail} p_q + \left( 1 - (1 - p_q)^k \right) \tilde{A}_q^{succ} p_q \right]. \tag{18}$$

*Subtracting,*

$$\Delta^{sel} - \Delta^{all} = -\eta \sum_{q \in U} \left[ 1 - (1 - p_q)^k \right] \left( \tilde{A}_q^{fail} - \tilde{A}_q^{succ} \right) p_q. \tag{19}$$

*Each term in the sum is $\leq 0$ since the bracketed term is positive and $\tilde{A}_q^{fail} \geq \tilde{A}_q^{succ}$. Therefore, $\Delta^{sel} \geq \Delta^{all}$.*

## B.3 MAIN THEOREM

**Guide-GRPO improves learning efficiency** Let $U$ be the set of prompts $q$ unsolved by the current policy $\pi_\theta$. Suppose that, in expectation over unsolved prompts and the group $G_q$ of guided and unguided trajectories, the guided advantage is positive:

$$\mathbb{E}_{q \in U} \left[ \mathbb{E}_{y \sim \pi_\theta(\cdot | \tilde{x}_q)} \left[ \tilde{A}_q(y; G_q) \right] \right] > 0.$$

Then for all $\eta$ sufficiently small, the one-step expected improvement, $\Delta \mathcal{R}$, under Guide-GRPO exceeds that of Vanilla GRPO, to first order in $\eta$:

$$\mathbb{E}[\Delta \mathcal{R}_{Guide}] > \mathbb{E}[\Delta \mathcal{R}_{Vanilla}], \tag{20}$$

where

$$\mathbb{E}[\Delta \mathcal{R}_{Vanilla}] = \eta \sum_{q \in U} A_q p_q^2 + \mathcal{O}(\eta^2), \tag{21}$$

$$\mathbb{E}[\Delta \mathcal{R}_{Guide}] = \eta \sum_{q \in U} \left[ A_q p_q^2 + (1 - p_q)^k \mathbb{E}_{y \sim \pi_\theta(\cdot | \tilde{x}_q)}[\tilde{A}_q(y)] p_q \right] + \mathcal{O}(\eta^2), \tag{22}$$

and $p_q = \mathbb{P}_{y \sim \pi_\theta(\cdot | x_q)} \left[ f(y) = y_q^* \right]$ denotes the success probability under the unguided policy.

**Proof 2** *Let $\theta_{t+1} = \theta_t + \eta \, g(\theta_t)$ for a small step size $\eta > 0$. We perform a first-order Taylor expansion:*

$$p_q(\theta_{t+1}) = p_q(\theta_t + \eta \, g) \tag{23}$$

$$p_q(\theta_t + \eta \, g) = p_q(\theta_t) + \eta \, \langle \nabla_\theta p_q(\theta_t), \, g \rangle + \mathcal{O}(\eta^2) \tag{24}$$

$$= p_q + \eta \, \langle \nabla_\theta p_q, \, g \rangle + \mathcal{O}(\eta^2). \tag{25}$$

*Using the log derivative trick:*

$$\nabla_\theta p_q = \mathbb{E}_{y \sim \pi_\theta(\cdot | x_q)} \left[ \mathbb{I}[f(y) = y_q^*] \nabla_\theta \log \pi_\theta(y \mid x_q) \right] \tag{26}$$

*which for a single trajectory entails:*

$$p_q(\theta_{t+1}) = p_q + \eta \, p_q \, \langle \nabla_\theta \log \pi_\theta(y_q \mid x_q), g \rangle + \mathcal{O}(\eta^2). \tag{27}$$

*Now substitute in the Guide-GRPO update:*

$$g = \sum_{q \in U} A_q \nabla_\theta \log \pi_\theta(y_q \mid x_q) + \sum_{q \in U} (1 - p_q)^k \tilde{A}_q \nabla_\theta \log \pi_\theta(y_q \mid x_q), \tag{28}$$

$$= \sum_{q \in U} \left[ A_q + (1 - p_q)^k \tilde{A}_q \right] \nabla_\theta \log \pi_\theta(y_q \mid x_q). \tag{29}$$

*Then:*

$$\langle \nabla_\theta \log \pi_\theta(y_q \mid x_q), g \rangle = \left[ A_q + (1 - p_q)^k \tilde{A}_q \right] \| \nabla_\theta \log \pi_\theta(y_q \mid x_q) \|^2. \tag{30}$$

*Therefore:*

$$p_q(\theta_{t+1}) - p_q = \eta \, p_q \left[ A_q + (1 - p_q)^k \tilde{A}_q \right] \| \nabla_\theta \log \pi_\theta(y_q \mid x_q) \|^2 + \mathcal{O}(\eta^2). \tag{31}$$

*Summing over q:*

$$\mathbb{E}[\Delta \mathcal{R}_{Guide}] = \sum_{q \in U} \mathbb{E}[p_q(\theta_{t+1}) - p_q(\theta_t)] \tag{32}$$

$$= \eta \sum_{q \in U} p_q \left[ A_q + (1 - p_q)^k \mathbb{E}[\tilde{A}_q] \right] \|\nabla_\theta \log \pi_\theta(y_q \mid x_q)\|^2 + \mathcal{O}(\eta^2). \tag{33}$$

*Compare to vanilla:*

$$\mathbb{E}[\Delta \mathcal{R}_{Vanilla}] = \eta \sum_{q \in U} A_q \, p_q \, \|\nabla_\theta \log \pi_\theta(y_q \mid x_q)\|^2 + \mathcal{O}(\eta^2). \tag{34}$$

*As long as $\mathbb{E}_{q \in U} \left[ \mathbb{E}_{y \sim \pi_\theta(\cdot \mid \tilde{x}_q)} \left[ \tilde{A}_q(y; G_q) \right] \right] > 0$, the guide update yields a greater gain.*

**Interpretation**  Guide enables gradient updates on prompts where *all* plain rollouts fail, scaling its advantage by $(1 - p_q)^k$. Its relative gain over vanilla GRPO is therefore proportional to

$$(1 - p_q)^k \mathbb{E}_{y \sim \pi_\theta(\cdot \mid \tilde{x}_q)} \left[ \tilde{A}_q(y; G_q) \right] p_q, \tag{35}$$

which increases when

- *failure probability* $(1 - p_q)^k$ is large (hard prompts),

- *guided advantage* $\mathbb{E}_{y \sim \pi_\theta(\cdot \mid \tilde{x}_q)}[\tilde{A}_q(y; G_q)]$ is large on average relative to the full rollout group,

- the baseline success probability $p_q$ is non-zero (so credit can propagate).

## C  EFFECTIVE VS ABSOLUTE CAPABILITY GAIN

We can set $k$ equal to the number of rollouts per problem used during training, which we call *effective* capability gain, or to the convergence of pass@$k$ curves where each additional sample provides a relative pass@$k$ improvement below some threshold $\epsilon$, which we define as *absolute* capability gain.

Let $k_{\text{eff}}$ be the number of rollouts per problem used in RLVR *training*. We define the **effective capability gain** as

$$\mathcal{G}_{\text{eff}} = \sum_{i \in U} \mathbb{I}\left[ \forall \hat{y} \in \mathcal{Y}_i^{(k_{\text{train}})} : \hat{y} \neq y_i \ \wedge \ \hat{y}_i^{\pi_{\text{RL}}} = y_i \right]. \tag{36}$$

In contrast, we can define absolute capability gain. Let

$$k_{\text{abs}} = \min\left\{ k : \frac{\text{pass@}k - \text{pass@}(k-1)}{\text{pass@}(k-1)} < \epsilon \right\}$$

be the smallest sample size at which additional rollouts yield $< \epsilon$ relative improvement. Then the **absolute capability gain** is

$$\mathcal{G}_{\text{abs}} = \sum_{i \in U} \mathbb{I}\left[ \forall \hat{y} \in \mathcal{Y}_i^{(k_{\text{abs}})} : \hat{y} \neq y_i \ \wedge \ \hat{y}_i^{\pi_{\text{RL}}} = y_i \right]. \tag{37}$$

## D    TRAINING DYNAMICS

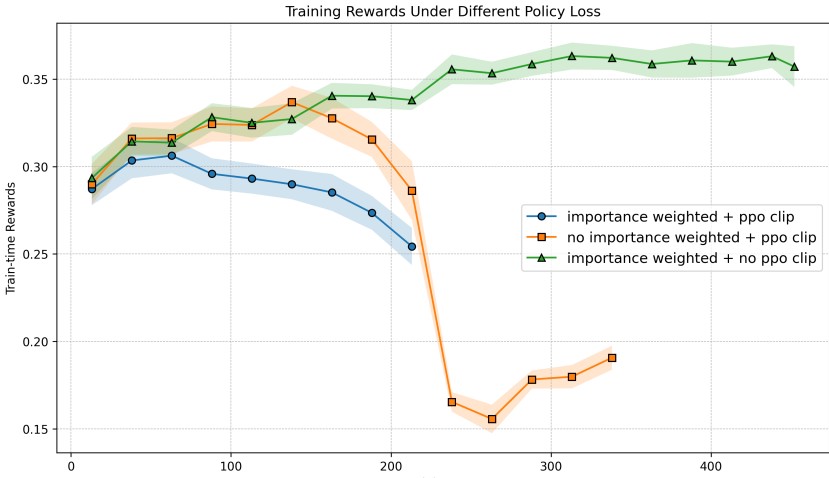

Figure 4: Comparison of train-time rewards under different policy loss computation when training with guided trajectories. The most stable training setup is when the importance weight is considered relative to the sampling distribution (prompt + guidance in context) and the typical ppo clip in the probability ratios is removed.

# E    GUIDED TRAJECTORY POLICY RESHAPING

Inspired by Yan et al. (40), we introduce a *policy-reshaping factor* that re-weights gradient contributions from off-policy (guided) samples, with the goal of amplifying learning signals from low-probability tokens in guided rollouts.

Let $w_i$ be the importance-weight ratio (as defined in Equation 2.2) for token $i$ in a guided trajectory $\mathbf{x}$, and let $\mathbf{w} = \{w_1, \ldots, w_{|\mathbf{x}|}\}$. While Yan et al. apply the static transform $f(x) = 1/(1 + x)$, we propose an **adaptive** alternative:

$$f(w_i) \;=\; \frac{1}{P_{90}(\mathbf{w}) + w_i},$$

where $P_{90}(\mathbf{w})$ denotes the 90th percentile of the ratios in $\mathbf{w}$. This normalisation boosts the relative contribution of tokens whose ratios fall below the percentile threshold, while tempering the influence of outliers with extremely large ratios. Because $P_{90}(\mathbf{w})$ is recomputed for every rollout, the reshaping adapts automatically to changes in the distributional gap between guided and unguided sampling throughout training. We experimented with 70, 80, and 90th percentiles and found that 90th percentile led to the best performance.

Preliminary experiments show modest but consistent gains on our validation benchmarks. Nevertheless, a systematic ablation (e.g. exploring different percentile thresholds or coupling the factor with temperature-scaled guidance) remains future work. We therefore present these results here in the Appendix for completeness rather than as a conclusive endorsement of the method.

# F    FULL PERFORMANCE RESULTS

Table 4: Pass@1 with greedy decoding and Pass@16 with temperature 1.0 performance with 95% confidence intervals. The CI for pass@1 is captured from 5 indepedent runs and the CI for pass@16 is captured from the variance of the 16 samples per prompt.

| Dataset | Metric | Guide-GRPO | Filter-GRPO | GRPO | SFT | Base |
|---------|--------|------------|-------------|------|-----|------|
| MATH500 | P@1 | 82.68 ± (0.10) | 80.80 ± (0.00) | 79.00 ± (0.00) | 72.80 ± (0.00) | 68.80 ± (0.00) |
|         | P@16 | 93.60 ± (0.54) | 92.60 ± (0.57) | 90.80 ± (0.63) | 87.00 ± (0.74) | 89.60 ± (0.67) |
| GSM8K   | P@1 | 91.43 ± (0.12) | 91.84 ± (0.04) | 91.71 ± (0.04) | 88.22 ± (0.06) | 83.21 ± (0.14) |
|         | P@16 | 97.73 ± (0.20) | 96.59 ± (0.24) | 96.97 ± (0.23) | 96.89 ± (0.23) | 97.19 ± (0.22) |
| MINERVA | P@1 | 32.35 ± (0.14) | 30.51 ± (0.00) | 32.72 ± (0.00) | 26.25 ± (0.18) | 26.47 ± (0.00) |
|         | P@16 | 47.43 ± (1.48) | 43.75 ± (1.47) | 45.96 ± (1.48) | 35.29 ± (1.42) | 43.01 ± (1.47) |
| OLYMPIAD | P@1 | 43.11 ± (0.22) | 40.21 ± (0.12) | 39.56 ± (0.00) | 35.05 ± (0.28) | 33.07 ± (0.11) |
|         | P@16 | 64.59 ± (0.90) | 60.89 ± (0.92) | 61.33 ± (0.92) | 50.52 ± (0.94) | 52.15 ± (0.94) |
| AMC     | P@1 | 63.61 ± (0.88) | 60.24 ± (0.00) | 62.41 ± (0.47) | 50.36 ± (0.88) | 47.47 ± (0.58) |
|         | P@16 | 84.34 ± (1.95) | 84.34 ± (1.95) | 84.34 ± (1.95) | 69.88 ± (2.47) | 78.31 ± (2.22) |
| AIME 24 | P@1 | 30.67 ± (1.31) | 18.67 ± (1.60) | 26.67 ± (0.00) | 11.33 ± (3.92) | 6.67 ± (0.00) |
|         | P@16 | 56.67 ± (4.43) | 53.33 ± (4.46) | 60.00 ± (4.38) | 23.33 ± (3.78) | 33.33 ± (4.22) |
| AIME 25 | P@1 | 13.33 ± (0.00) | 13.33 ± (0.00) | 13.33 ± (0.00) | 6.67 ± (0.00) | 3.33 ± (0.00) |
|         | P@16 | 46.67 ± (4.46) | 33.33 ± (4.22) | 30.00 ± (4.10) | 13.33 ± (3.04) | 13.33 ± (3.04) |
| Macro Avg. | P@1 | 51.03 ± (0.40) | 47.94 ± (0.25) | 49.34 ± (0.07) | 40.42 ± (0.83) | 38.43 ± (0.12) |
|         | P@16 | 70.15 ± (2.00) | 66.40 ± (1.98) | 67.06 ± (1.96) | 53.75 ± (1.80) | 58.13 ± (1.83) |
| Micro Avg. | P@1 | 70.66 ± (0.00) | 69.76 ± (0.00) | 69.59 ± (0.00) | 63.80 ± (0.00) | 61.16 ± (0.00) |
|         | P@16 | 83.29 ± (0.34) | 81.23 ± (0.35) | 81.44 ± (0.35) | 76.28 ± (0.39) | 78.31 ± (0.37) |

Table 5: Pass@1 with greedy decoding and Pass@16 with temperature 1.0 performance with 95% confidence intervals. The CI for pass@1 is captured from 5 indepedent runs and the CI for pass@16 is captured from the variance of the 16 samples per prompt.

| Benchmark | Metric | Guide-32B-8K | GRPO-32B-8K | Guide-7B-8K | GRPO-7B-8K |
|---|---|---|---|---|---|
| MATH500 | P@1 | **85.08** ± (0.20) | 84.00 ± (0.12) | 83.08 ± (0.16) | 79.96 ± (0.08) |
| | P@16 | 94.80 ± (0.49) | **95.20** ± (0.47) | 94.60 ± (0.50) | 94.40 ± (0.50) |
| GSM8K | P@1 | **95.60** ± (0.05) | 95.60 ± (0.00) | 92.25 ± (0.06) | 91.46 ± (0.04) |
| | P@16 | **98.03** ± (0.19) | 97.88 ± (0.19) | 97.80 ± (0.20) | 98.18 ± (0.18) |
| MINERVA | P@1 | **35.59** ± (0.18) | 35.51 ± (0.18) | 29.41 ± (0.00) | 30.15 ± (0.00) |
| | P@16 | **50.00** ± (1.49) | 49.63 ± (1.49) | 49.26 ± (1.49) | 48.53 ± (1.48) |
| OLYMPIAD | P@1 | **54.52** ± (0.44) | 48.06 ± (0.07) | 43.67 ± (0.17) | 42.37 ± (0.22) |
| | P@16 | **71.11** ± (0.85) | 68.89 ± (0.87) | 66.37 ± (0.89) | 65.33 ± (0.90) |
| AMC | P@1 | **65.06** ± (0.88) | 63.61 ± (0.75) | 62.65 ± (0.00) | 53.73 ± (0.58) |
| | P@16 | **87.95** ± (1.75) | **87.95** ± (1.75) | 90.36 ± (1.59) | 89.16 ± (1.67) |
| AIME 24 | P@1 | **32.67** ± (1.31) | 20.00 ± (0.00) | 16.67 ± (0.00) | 26.67 ± (0.00) |
| | P@16 | **66.67** ± (4.38) | 60.00 ± (4.22) | 63.33 ± (4.31) | 53.33 ± (4.46) |
| AIME 25 | P@1 | **25.33** ± (3.33) | 23.33 ± (0.00) | 19.33 ± (1.31) | 20.67 ± (1.31) |
| | P@16 | **50.00** ± (4.43) | 43.33 ± (4.43) | 43.33 ± (4.43) | 30.00 ± (4.10) |
| Macro Avg. | P@1 | **56.26** ± (0.91) | 52.87 ± (0.16) | 49.58 ± (0.24) | 49.29 ± (0.32) |
| | P@16 | **74.08** ± (1.94) | 71.84 ± (1.92) | 72.15 ± (1.91) | 68.42 ± (1.90) |
| Micro Avg. | P@1 | **76.36** ± (0.00) | 74.47 ± (0.00) | 71.15 ± (0.00) | 69.89 ± (0.00) |
| | P@16 | **85.63** ± (0.32) | 84.94 ± (0.32) | 84.29 ± (0.33) | 83.84 ± (0.33) |

Table 6: Comparison of Pass@1 (greedy decoding) and Pass@16 (temperature 1.0) performance on several math benchmarks across different training algorithms using Llama 3.1 8B Instruct

| Benchmark | Metric | Guide-GRPO | Filter-GRPO | Base |
|---|---|---|---|---|
| MATH500 | P@1 | 60.60 | 53.60 | 47.96 |
| | P@16 | 81.20 | 73.40 | 79.80 |
| GSM8K | P@1 | 84.61 | 83.47 | 84.22 |
| | P@16 | 96.74 | 94.69 | 97.12 |
| MINERVA | P@1 | 26.74 | 23.52 | 20.61 |
| | P@16 | 44.85 | 37.50 | 45.22 |
| OLYMPIAD | P@1 | 24.31 | 20.24 | 16.31 |
| | P@16 | 40.00 | 37.04 | 36.74 |
| AMC | P@1 | 24.89 | 23.86 | 23.61 |
| | P@16 | 45.78 | 48.19 | 45.42 |
| AIME 24 | P@1 | 6.67 | 0.00 | 3.33 |
| | P@16 | 23.33 | 20.00 | 23.33 |
| AIME 25 | P@1 | 0.00 | 0.00 | 0.00 |
| | P@16 | 10.00 | 10.00 | 0.00 |
| Macro Avg. | P@1 | 32.54 | 29.24 | 28.00 |
| | P@16 | 48.84 | 45.83 | 46.80 |
| Micro Avg. | P@1 | 57.68 | 54.60 | 52.81 |
| | P@16 | 72.95 | 69.34 | 72.04 |

# G  GUIDANCE GENERATION

To generate prompt-specific guidance, we used the prompt below at temperature 0 with GPT-4o.

```
'''
You are an expert math tutor with years of experience helping students
↪ understand difficult concepts without solving problems for them. Your
↪ task is to analyze a math problem and its reference solution, then
↪ create a series of helpful hints that guide the student toward
↪ discovering the solution independently.

[question]: {question}

[reference_answer]: {reference_answer}

When creating hints, follow these principles:
- Start with conceptual hints that point to relevant mathematical
↪ principles
- Progress to more specific strategic hints about approach
- Offer guidance on potential roadblocks without revealing key steps
- Never provide direct calculations or formulas that would bypass the
↪ student's own reasoning
- Include 3-5 hints in ascending order of directness

[guidance]:
1. [Conceptual hint focusing on relevant mathematical principles]
2. [Strategic hint about approach without revealing steps]
3. [Hint addressing common misconception or roadblock]
4. [More direct hint if needed]
5. [Final hint that guides toward the solution method without providing
↪ it]
'''
```

The following are some examples on the guidance generated using the prompt above:

---

### Example A: (Algebra)

User: Find $\frac{a^{12}-4096}{64 a^{6}}$, if $\frac{a^{2}}{4}-\frac{4}{a^{2}}=3$.

A hint to the problem is provided below:

```
[HINT_START]
1. Notice that the given condition and the expression both involve terms of the form a²
↪  and its reciprocal. Think about how the identity (a²/4) - (4/a²) might be used as a
↪  building block for factoring the larger expression.
2. Consider rewriting the numerator in a way that reveals a common factor with the given
↪  condition. Ask yourself if a difference of powers or a recognizable factorization
↪  pattern might help connect the two parts of the expression.
3. It might be useful to make a substitution, such as letting x = a²/4, to help simplify
↪  both the given equation and the overall expression. How can this substitution
↪  simplify the structure of the problem?
4. Once you have expressed the terms in a unified form using your substitution, think
↪  about how to factor or combine terms so that the given condition naturally appears in
↪  the algebraic manipulation.
5. Finally, reflect on any familiar algebraic identities or factorizations you know that
↪  involve higher powers; these might help you write the expression in a form that can
↪  be directly evaluated using the given value. Use these observations to guide your
↪  step-by-step approach toward the final simplified result.
[HINT_END]
```

---

### Example B: (Geometry)

User:  The area of a new playground is square and measures $1600 \mathrm{~m}^{2}$. How
↪  long is one side of the playground? How many meters of fence are needed for three
↪  sides?

A hint to the problem is provided below:

```
[HINT_START]
1. Think about what it means for a shape to be a square, and how its area relates to the
↪  length of one of its sides.
2. Consider how you might determine the side length from the given area without directly
↪  calculating it--what operation relates area and side length for a square?
3. Once you have an idea for finding the side length, think about how you would compute
↪  the total length of fence needed for three sides of a square playground. What does
↪  that imply about multiplying the side length?
4. Be cautious about mixing up the operations needed to go from area to side length and
↪  then to the fence perimeter. Ensure each step follows from the properties of a
↪  square.
5. Finally, try to translate the relationships into a step-by-step approach that first
↪  finds the side of the square and then uses that to calculate the overall fence length
↪  for three of its sides.
[HINT_END]
```

Table 7: Two sample hint blocks produced by our pipeline.

**Example C: Effect of Hints on Model Reasoning**

**(1) Question (no hints)**

```
In the right-angled triangle $ABC$, the lengths of the legs $AC=3$ and $BC=4$. Point $A$
↪  was moved to point $A_{1}$ parallel to $BC$, then point $B$ was moved to point
↪  $B_{1}$ parallel to $A_{1}C$, and finally, point $C$ was moved to point $C_{1}$
↪  parallel to $A_{1}B_{1}$, such that the resulting triangle $A_{1}B_{1}C_{1}$ is
↪  right-angled at $B_{1}$, and the length of the leg $A_{1}B_{1}$ is 1 unit. What is
↪  the length of the leg $B_{1}C_{1}$?'
```

**(2) Model response without hints (incorrect)**

```
To solve this problem, we need to understand the geometric transformations and the
↪  properties of the right-angled triangles involved. Let's break down the problem
↪  step-by-step.\n\n1. **Initial Triangle \\(ABC\\)**:\n   - \\(AC = 3\\)\n   - \\(BC =
↪  4\\)\n   - \\(AB = \\sqrt{AC^2 + BC^2} = \\sqrt{3^2 + 4^2} = 5\\)\n\n2.
↪  **Transformation of Points**:\n   - Point \\(A\\) is moved to \\(A_1\\) parallel to
↪  \\(BC\\).\n   - Point \\(B\\) is moved to \\(B_1\\) parallel to \\(A_1C\\).\n   -
↪  Point \\(C\\) is moved to \\(C_1\\) parallel to \\(A_1B_1\\).\n\n3. **Resulting
↪  Triangle \\(A_1B_1C_1\\)**:\n   - The triangle \\(A_1B_1C_1\\) is right-angled at
↪  \\(B_1\\).\n   - The length of the leg \\(A_1B_1 = 1\\).\n\n\hl{Since the points are
↪  moved parallel to the sides of the triangle, the resulting triangle} \\(A_1B_1C_1\\)
↪  is similar to the original triangle \\(ABC\\). The ratio of the corresponding sides
↪  of the triangles will be the same.\n\nLet's denote the length of the leg \\(B_1C_1\\)
↪  as \\(x\\). Since the triangles are similar, the ratio of the corresponding sides
↪  will be the same. The ratio of the sides of the triangles \\(A_1B_1C_1\\) and
↪  \\(ABC\\) is \\(1/5\\).\n\nSo, the length of the leg \\(B_1C_1\\) will be \\(4 \\times
↪  (1/5) = 4/5\\).\n\nLet's confirm this with Python and sympy.\n\n```python\nimport
↪  sympy as sp\n\n# Given lengths of the original triangle\nAC = 3\nBC = 4\nAB =
↪  sp.sqrt(AC**2 + BC**2)\n\n# The length of the leg A1B1 is given as 1\nA1B1 = 1\n\n#
↪  The ratio of the sides of the similar triangles\nratio = A1B1 / AB\n\n# The length of
↪  the leg B1C1\nB1C1 = BC * ratio\n\nprint(B1C1)\n```\n```output\n4/5\n```\nThe length
↪  of the leg \\(B_1C_1\\) in the resulting triangle \\(A_1B_1C_1\\) is
↪  \\(\\boxed{\\frac{4}{5}}\\).
```

**(3) Question (with hints)**

```
In the right-angled triangle $ABC$, the lengths of the legs $AC=3$ and $BC=4$. Point $A$
↪  was moved to point $A_{1}$ parallel to $BC$, then point $B$ was moved to point
↪  $B_{1}$ parallel to $A_{1}C$, and finally, point $C$ was moved to point $C_{1}$
↪  parallel to $A_{1}B_{1}$, such that the resulting triangle $A_{1}B_{1}C_{1}$ is
↪  right-angled at $B_{1}$, and the length of the leg $A_{1}B_{1}$ is 1 unit. What is
↪  the length of the leg $B_{1}C_{1}$? A hint to the problem is provided below:
↪  [HINT_START] 1. Consider that moving points along lines parallel to a side of a
↪  triangle does not change the distance (or altitude) from that side, so the area
↪  remains constant from the original triangle to the resulting triangle. 2. Remember
↪  that for a right triangle the area can be found using half the product of its two
↪  legs. With one leg given as 1, think about how you can relate this familiar formula
↪  to the unchanged area of the triangle. 3. A common pitfall is to assume that moving
↪  the points alters side ratios or angle measures aside from the given right angle;
↪  keep in mind that the key is the invariant area. 4. Now, equate the area expressed in
↪  the original triangle with that expressed in the new right triangle, and set up an
↪  equation that will allow you to solve for the unknown leg length without doing
↪  detailed arithmetic. 5. Finally, confirm that the relationship you've set up is
↪  consistent with how translation preserves distances from the parallel lines--this
↪  should guide you to finding the missing leg length. [HINT_END] Consider the hint but
↪  start your solution from scratch and do not directly reference the hint (e.g. avoid
↪  phrases like "using the hint").
```

**(4) Model response with hints (correct)**

```
To solve this problem, we need to use the fact that the area of the triangle remains
↪  constant before and after the movements of the points. Let's break down the problem
↪  step-by-step:\n\n1. **Initial Triangle \(ABC\)**:\n - \(AC = 3\)\n - \(BC = 4\)\n -
↪  The area of triangle \(ABC\) is \(\frac{1}{2} \times AC \times BC = \frac{1}{2}
↪  \times 3 \times 4 = 6\).\n\n2. **Final Triangle \(A_1B_1C_1\)**:\n - \(A_1B_1 = 1\)\n
↪  - Let \(B_1C_1 = x\)\n - The area of triangle \(A_1B_1C_1\) is \(\frac{1}{2} \times
↪  A_1B_1 \times B_1C_1 = \frac{1}{2} \times 1 \times x = \frac{x}{2}\).\n\nSince the
↪  area remains constant, we have:\n\[\n6 = \frac{x}{2}\n\]\nSolving for \(x\), we
↪  get:\n\[\nx = 12\n\]\n\nSo, the length of the leg \(B_1C_1\) is \(12\).\n\nLet's
↪  confirm this with Python code to ensure the result is accurate.\n```python\n# Given
↪  values for the initial triangle ABC\nAC = 3\nBC = 4\n\n# Calculate the area of the
↪  initial triangle ABC\narea_ABC = 0.5 * AC * BC\n\n# Given value for the final
↪  triangle A1B1C1\nA1B1 = 1\n\n# Let B1C1 be x\n# The area of the final triangle A1B1C1
↪  is 0.5 * A1B1 * B1C1\n# Since the area remains constant, we have:\n# area_ABC = 0.5 *
↪  A1B1 * B1C1\n# 6 = 0.5 * 1 * x\n# x = 12\n\n# Calculate the length of B1C1\nB1C1 = 2 *
↪  area_ABC / A1B1\nprint(B1C1)\n```\n```output\n12.0\n```\nThe length of the leg
↪  \(B_1C_1\) is \(\boxed{12}\).
```

Table 8: Example where the model's initial solution (without hints) is incorrect; after receiving guidance, the model corrects its reasoning and produces the right answer. Without guidance, the model incorrectly assumes that the three parallel translations produce complete triangle similarity, adopts a side-length ratio of 1:5, and consequently predicts $B_1C_1 = 4 \times \frac{1}{5} = \frac{4}{5}$. After the area-

# H TRAINING DETAILS: REINFORCEMENT LEARNING WITH VERIFIABLE REWARDS (RLVR) TRAINING

Prior to RLVR training, we perform one epoch of supervised fine-tuning (SFT) using the AMPS dataset (15) on all Qwen 2.5 base models (13) to ensure that the models produce reasoning-formatted outputs. The prompt used for both RLVR and SFT is shown below:

```
'''
A conversation between user and assistant.
The user asks a question, and the assistant solves it.
The assistant first thinks about the reasoning process in the mind and
↪   then provides the user with the answer.
The reasoning process and answer are enclosed within <think> </think> and
↪   <answer> </answer> tags, respectively, i.e., <think> reasoning
↪   process here </think> <answer> answer here </answer>.

User: {{question}}

Assistant:
'''
```

For RLVR, we gathered the publicly available verifiable rewards dataset into three broad splits.

Table 9: Training dataset composition for RLVR

| Dataset | Number of Examples |
|---------|-------------------|
| Math | 450,000 |
| Code | 25,276 |
| STEM | 38,958 |
| Total | 514,234 |

For training, we used the following hyperparameters when running the open-source VeRL (41) package with GRPO:

Table 10: Hyper-parameters for GRPO training

| hyperparameter and settings | value |
|-----------------------------|-------|
| train batch size | 1024 |
| ppo mini batch | 512 |
| number rollouts per prompt | 8 |
| training steps | 256 |
| actor learning rate | 1e-6 |
| kl coeff | 0 |
| entropy coeff | 0 |
| prompt max length | 1024 |
| generation max length | 3072 |
| policy model temperature | 1 |
| optimizer | Adam |

For SFT, we used the following hyperparameters,

Table 11: Hyper-parameters for SFT training

| hyperparameter and settings | value |
| --- | --- |
| train batch size | 32 |
| training epochs | 1 |
| actor learning rate | 1e-5 |
| max context length | 4092 |
| optimizer | Adam |

# I  GUIDE-GRPO TOWARDS MATHEMATICAL REASONING DETAILS

In this appendix, we discuss further experimental details for our results in 3.2.

## I.1  GUIDANCE PASS@K ON TRAINING DATA

**Dataset and Model Details**   For the evaluation, we randomly sample 10k samples from OpenR1-Math-220k (24) train subset and use the Qwen-2.5-Math-7B trained by (40) for running inference. For these prompts we generate guidance using reference solution by using the prompt described in Appendix G.

**Capability Gain and Distillation**   We compute the capability gain (C) and distillation set (D) from the model rollouts without guidance using the strategy described in Section 3.1. For each set, we then measure pass @ k for the guided and non-guided model with samples generated with temperature 1.0. We see that the model with guidance can solve more unsolvable questions in comparsion to the non guided model and the number of questions also increases with rising k. Similarly, the guided model also shows significantly higher pass @ 1 with a rising trend with increasing k.

## I.2  GRPO AND SFT TRAINING

**Training Data**   For the training data, we use the default subset of OpenR1-Math-220k, which contains 93.7K math reasoning tasks that have been sourced for several math competitions, textbooks and online forums. We use the prompt, the final answer, and the reference solution, which is the solution that is scraped from the corresponding source, not the llm-generated solution. In order to format the training data, we leverage the system prompt used by Yan et al. that encourages the model to first think through the problem and then provide an answer in boxed format (40):

```
Your task is to follow a systematic, thorough reasoning process before
↪   providing the final solution. This involves analyzing, summarizing,
↪   exploring, reassessing, and refining your thought process through
↪   multiple iterations. Structure your response into two sections:
↪   Thought and Solution. In the Thought section, present your reasoning
↪   using the format: \"<think>\n {thoughts} </think>\n\". Each thought
↪   should include detailed analysis, brainstorming, verification, and
↪   refinement of ideas. After \"</think>\n,\" in the Solution section,
↪   provide the final, logical, and accurate answer, clearly derived from
↪   the exploration in the Thought section. If applicable, include the
↪   answer in \\boxed{} for closed-form results like multiple choices or
↪   mathematical solutions.
```

Using this system prompt, we simply apply the chat template with the user prompt to formulate the training prompt for GRPO training.

**Training hyperparameter**   Table 12 contains the common training hyper-parameter for training the policy models under vanilla GRPO and Guide-GRPO and Table 13 contains the training hyper-parameters for the SFT training on reference solutions:

| hyperparameter and settings | value |
| --- | --- |
| train batch size | 1024 |
| ppo mini batch | 512 |
| number rollouts per prompt | 8 |
| training epochs | 2 |
| actor learning rate | 1e-6 |
| kl coeff | 0 |
| entropy coeff | 0 |
| prompt max length | 1024 |
| generation max length | 3072 |
| policy model temperature | 1 |
| optimizer | Adam |

Table 12: Hyper-parameters for GRPO training

| hyperparameter and settings | value |
| --- | --- |
| train batch size | 32 |
| training epochs | 2 |
| actor learning rate | 1e-5 |
| max context length | 4092 |
| optimizer | Adam |

Table 13: Hyper-parameters for SFT training

**Training Setup**   We fork the open-sourced VeRL (41) training package. We make the following modifications to the code for implementing Guide-GRPO:

- Implement filtering of prompts for which all solution trajectories are all correct or all incorrect
- Adjusted importance weighting to calculate log probabilities relative to prompt plus guidance
- Update the data-loaders to include the token ids for prompt with guidance
- Dynamic re-rolls from prompt groups for which all trajectories are incorrect using prompt plus guidance in the context of the policy model

**Compute**   All GRPO model training used 2 nodes for a total of 16 gpus with 88 CPU cores, 80 Gi GPU memory and 1.5TB system memory per node. The total GRPO training time ranged between 36 to 48 hours. The SFT training was done on 1 node with training time of approximately 6 hours.

