# OpenReview forum: "Adaptive Guidance Accelerates Reinforcement Learning of Reasoning Models"
_ICLR.cc/2026/Conference — ICLR 2026 Conference Withdrawn Submission_

### Official Review · Reviewer_h1a7 · 2025-10-20

**Soundness:** 1
**Presentation:** 1
**Contribution:** 1
**Rating:** 2
**Confidence:** 3

**Summary:**

This paper studies how some reasoning models improve performance during reinforcement learning.

**Strengths:**

The paper offers a potentially new insight in that the observed performance gains in reinforcement learning is largely via the reduction in variance in solving the set of solvable problems, rather than expanding the set of solvable problems.     However, the validity and value of this insight across the paper is very hard to estimate owing to a number of confounding factors as below.

**Weaknesses:**

Paper is poorly written with undefined quantities and prone to excessive notation.   Even the abstract seems to contradict itself, e.g. lines 15-17
"We find that while capability gain exists across model scales, learning to solve new problems is primarily driven through self-distillation."
By the papers own definition, "self-distillation" does not solve new problems.

As another example, the quantities $y_i, \hat{y}_i$ that appear repeatedly in the paper starting with Equation (1) are undefined.    If one assumes that $y_i$ is the output of the LLM on a prompt $x_i$, is $y_i$ uniquely determined or a random variable?     Or is $y_i$ the "correct answer" to prompt $x_i$.   What if the correct answer is not unique?

The reader is left to decipher that the undefined notation $\mathbb{I}$ might be the indicator function.  Likewise "policy $\pi_{RL}$" is used without definition on line 85.

Lines 91-92: "where $U^{\pi_{init}}$ is the set of indices of unsolved problems prior to RL and $S^{\pi_{init}}$ is the set of indices of solved problems prior to RL. We define solved and unsolved here via pass@1 correctness."
pass@1 is a random variable in the interval [0,1].   What does correctness mean?

With regard to the equation on line 99, the subscript $i$ in the summations seems to have multiple domains.

Overall, it is impossible to determine whether the theoretical results are valid or valuable.  In light of the above, it is difficult to determine what exactly is tested in the experiments.

**Questions:**

The primary insight of the paper on the reduction of variance aka self-distillation is interesting.  Rather than "proving" this under awkward conditions and heavy notational machinery, a clean rewrite of the paper focused on experiments might make strong contribution?

Is the citation format compressed in order to squeeze more into the page limit?

---

### Official Review · Reviewer_bjNe · 2025-10-29

**Soundness:** 2
**Presentation:** 3
**Contribution:** 2
**Rating:** 6
**Confidence:** 4

**Summary:**

This paper studies how RLVR improves reasoning in LLMs and shows that most gains come from self-distillation, compressing pass@k success into pass@1, rather than true capability growth. To accelerate learning on hard problems, the authors propose Guide, an adaptive RL method that adds natural-language hints only when all rollouts fail and corrects the updates with importance weighting. The resulting Guide-GRPO algorithm improves sample efficiency and generalization.

**Strengths:**

1. The paper evaluates the Guide algorithm not only through experiments but also provides a theoretical explanation for its effectiveness, which is commendable.

2. Some experiments in the paper are repeated multiple times to ensure the accuracy and reproducibility of the results.

3. The paper also compares multiple baselines, offering a comprehensive empirical evaluation.

**Weaknesses:**

1. The proposed Guide method appears somewhat trivial and lacks strong novelty. Essentially, the approach introduces hints generated by a stronger model (e.g., GPT-4o) to help the policy model solve difficult problems. This idea is conceptually similar to few-shot prompting or knowledge injection, where additional external information is provided to improve sample efficiency rather than fundamentally changing the learning paradigm.

2. In essence, Guide can be viewed as knowledge injection from a larger teacher model into the student policy during RL training. This resembles a cold-start + RL setup, where external supervision from a stronger model initializes or accelerates learning rather than achieving self-improvement purely through exploration.

3. All experiments are conducted only on Qwen models within the math domain. Prior works [1][2] have shown that Qwen exhibits domain-specific advantages in mathematical reasoning, which raises concerns about the generalization of the proposed method to other domains (e.g., coding, science, or open-ended reasoning) or models.

[1] Zeng, Weihao, et al. "Simplerl-zoo: Investigating and taming zero reinforcement learning for open base models in the wild." arXiv preprint arXiv:2503.18892 (2025).
[2] Shao, Rulin, et al. "Spurious rewards: Rethinking training signals in rlvr." arXiv preprint arXiv:2506.10947 (2025).

**Questions:**

1. In Line 206, could the authors clarify the rationale behind using Pass@16 as the metric in their analysis? Why was k = 16 chosen specifically? Please justify this choice.


2. In Table 1, Filter-GRPO performs very similarly to or even slightly worse than vanilla GRPO, which is surprising since Filter-GRPO is typically expected to improve both training efficiency and performance stability by removing uninformative rollouts. Could the authors explain why this expected improvement does not appear here?

---

### Official Review · Reviewer_mY9F · 2025-10-31

**Soundness:** 3
**Presentation:** 3
**Contribution:** 2
**Rating:** 4
**Confidence:** 4

**Summary:**

The paper introduces Guide, an algorithm that injects adaptive natural-language hints into training when all rollouts for a prompt fail, and corrects the resulting off-policy gradients via importance weighting. Guide-GRPO demonstrates consistent improvements over vanilla RLVR baselines, showing better generalization and sample efficiency.

**Strengths:**

- The proposed Guide algorithm is well-motivated and theoretically grounded.
- The paper includes useful insights into how guidance affects entropy, exploration, and convergence stability.

**Weaknesses:**

- Questionable interpretation of “capability gain.” As *Limits-of-RLVR* [1] **argues, capability gain almost vanishes when k increases; apparent gains at small k may simply reflect undersampling of the model’s inherent capability. The paper acknowledges this but still interprets small absolute gains as substantive learning. A few hundred samples can hardly be called “ineffective” since it is way smaller than the actual language space.
- Limited novelty relative to prior off-policy or guided RL methods. The idea of leveraging guided or off-policy trajectories is not entirely new. Prior works such as LUFFY [2] and BREAD [3] have explored similar concepts of mixing guided rollouts or hint-based supervision.

[1] Does reinforcement learning really incentivize reasoning capacity in LLMs beyond the base model?

[2] Learning to Reason under Off-Policy Guidance

[3] BREAD: Branched Rollouts from Expert Anchors Bridge SFT & RL for Reasoning

**Questions:**

See weakness

---

### Official Review · Reviewer_yv9z · 2025-10-31

**Soundness:** 3
**Presentation:** 3
**Contribution:** 3
**Rating:** 6
**Confidence:** 4

**Summary:**

This paper investigates how RLVR improves reasoning capabilities in large language models in math domain. It identifies two mechanisms underlying performance gains: self-distillation, where models convert multi-sample success (pass@k) into single-sample success (pass@1), and capability gain, where models learn to solve previously unsolvable problems. Empirical analysis shows that progress in RLVR is largely dominated by self-distillation rather than genuine capability expansion. Building on these insights, the authors introduce Guide, a new family of adaptive reinforcement learning algorithms that provide contextual guidance—natural-language hints—to the model only when all unguided rollouts fail.  The method enhances both pass@1 and pass@16 across datasets like GSM8K, AIME, and OlympiadBench.

**Strengths:**

1. Clear decomposition of RLVR gains into **self-distillation** (compressing pass@k→pass@1) vs **capability gain**, giving a precise lens to analyze “why RL works” for reasoning models. Strong, multi-scale study (0.5B→72B; >500k problems across math/science/code) and careful pass@k protocol; shows self-distillation dominates while capability gain still exists.

2. Solid ablations (guidance thresholds) and training-stability analysis (importance weighting, PPO-clip) support the design choices.

3. Authors report similar improvements on LLaMA-3.1-8B Instruct besides Qwen, supporting portability of the method across model families. Also include different methods such as GRPO and Filter-GRPO, different context length.

**Weaknesses:**

1. The authors seem to have only validated the effectiveness of GUIDE in the mathematical domain, but it would be desirable to verify its effectiveness in other domains such as science, code, etc. I saw that the authors seem to have conducted tests on HumanEval at lines 234-236, but I did not see the corresponding results.

2. The authors mention using GPT-4o to produce hints, but it is unclear whether this might produce hallucinations or factual errors. It would be preferable to have human verification, conducting statistics or case studies.

3. Have the authors used models other than GPT-4o to produce hints, such as models stronger or weaker than GPT-4o, to see whether the quality of hints changes and what the ultimate impact on RL performance would be.

**Questions:**

Please refer to weaknesses

---

### Official Review · Reviewer_WQKP · 2025-11-01

**Soundness:** 2
**Presentation:** 2
**Contribution:** 3
**Rating:** 2
**Confidence:** 3

**Summary:**

This paper studies reinforcement learning on verifiable rewards, finding RLVR boosts performance via pass@k to pass@1 compression and capability gain, with self-distillation dominating new problem learning across 0.5B-72B models. It proposes Guide (e.g., GuideGRPO), which injects hints for full-failure problems and adjusts off-policy importance sampling. On a series of math benchmarks, GuideGRPO improves 7B/32B models by up to 4% macro-average over vanilla counterparts.

**Strengths:**

- The idea discussed in the paper is interesting, exploring the specific reasons behind RLVR's performance improvement.
- Analyzing the source of capabilities by comparing Pass@k and Pass@1 is highly reasonable.

**Weaknesses:**

- The overall writing and narration of the paper are not clear. Based on my understanding, Section 2.1 should be an analysis of RLVR, which then transitions to the motivation for proposing Guide. However, the details and conclusions of Section 2.1 are placed in Section 3.1. Moreover, the Introduction Section only introduces the Guide in the third contribution of the last paragraph. I am not clear about the transition from the analysis of RLVR to the proposal of Guide. Sections 2.1 and 3.1 would be better placed together.
- I think the definition of self-distillation, from the initial state where a problem can be solved with pass@16 to the final state where it can be solved with pass@1, is perhaps not sufficient. Maybe the initial state should also include the condition that the problem cannot be solved with pass@1.
- What is the mixed multi-task dataset used in Section 3.1.1?

**Questions:**

- In Section 3.1.1, why are 100 samples generated for pass@1 as well, with one sample selected for evaluation?
- Can the analysis of the source of capabilities in RLVR be conducted by different domains? Cause that Qwen + Math may be a relatively special case.

---

### Note · Authors · 2025-11-26

I have read and agree with the venue's withdrawal policy on behalf of myself and my co-authors.